# Immune Checkpoint Inhibitors and Opioids in Patients with Solid Tumours: Is Their Association Safe? A Systematic Literature Review

**DOI:** 10.3390/healthcare11010116

**Published:** 2022-12-30

**Authors:** Massimiliano Cani, Paolo Bironzo, Ferdinando Garetto, Lucio Buffoni, Paolo Cotogni

**Affiliations:** 1Oncology Unit, Department of Oncology, University of Turin, S. Luigi Gonzaga Hospital, 10043 Orbassano, Italy; 2Medical Oncology Department, Humanitas Gradenigo, 10153 Turin, Italy; 3Department of Biomedical Sciences, Humanitas University, Via Rita Levi Montalcini, 10153 Turin, Italy; 4Cottolengo Hospice, Via Cesare Balbo 16, 10023 Chieri, Italy; 5Pain Management and Palliative Care, Department of Anesthesia, Intensive Care and Emergency, Molinette Hospital, University of Turin, Corso Bramante, 88/90, 10126 Turin, Italy

**Keywords:** immunotherapy, opioids, NSCLC, concomitant drugs

## Abstract

Background: Immune checkpoint inhibitors (ICIs) represent one of the most effective treatments for patients with cancer. As their activity relies on host immune system reactivity, the role of concomitant medications such as corticosteroids and antibiotics has been extensively evaluated. Preclinical data suggest that opioids may influence the immune system. Methods: a systematic literature revision was performed using specific keywords on the major search engines. Two authors analysed all the studies and provided a selection of the following inclusion and exclusion criteria, respectively: 1. data collection of patients older than 18 years old affected by solid tumours; 2. description of ICIs efficacy in terms of PFS, OS, TTF, and ORR; 3. concomitant ICIs-opioids treatment and 1. language different from English; 2. not pertinent analyses. Results: 523 studies were analysed, and 13 were selected and included in our series. A possible negative interaction between oral opioids and ICIs efficacy was observed. Most evidence was retrospective, and studies were heterogeneous. Conclusions: Even if oral opioids seem to impact negatively on ICIs efficacy in cancer patients, to date there is not sufficient evidence to avoid their prescription in this population.

## 1. Introduction

Given the wide spread of immune checkpoint inhibitors (ICIs) as a treatment for several tumours, the definition of possible interactions between ICIs and concomitant drugs has recently gained importance. While the negative interactions of antibiotics and corticosteroids are now well known [1,2,3], other medications are still under investigation. Indeed, some in vitro and in vivo experiments have demonstrated morphine receptors on neoplastic cells [4]. The activation of such receptors may have an impact on both tumour growth and metastatic spread potential. However, a meta-analysis evaluating animal studies concluded that there is no evidence that analgesic, including opioids, increases metastases occurrence [5]. Moreover, prospective studies on patients with cancer failed to show any association between opioid use and the risk of recurrence in breast and colorectal cancer patients [6,7,8]. At the same time, the influence of opioids on the immune system has been widely studied, with controversial results, especially in patients with cancer.

The mechanisms of opioids on the immune system or immune cells have been studied for over 40 years. Some opioids are associated with immunosuppressive effects, with a developing knowledge that morphine, fentanyl, buprenorphine, and methadone suppress innate immunity while having different effects on adaptive immunity. It is similarly apparent that specific opioids have immunostimulatory effects, some exhibit dual effects, and others have no immunomodulatory effect [9]. Indeed, Wybran et al. showed that morphine can reduce T-cell rosettes formation in vitro, an effect that could be reversed by naloxone administration [10]. Conversely, short-term morphine use has been shown to induce IL-2 and IL-6 expression, while chronic use enhances T-reg cell activity, reduces Th17 function and increases μ opioid receptor mRNA expression in T lymphocytes [11,12,13,14]. In vitro studies also showed that morphine reduces T-helpers 1 activation while increasing T-helpers 2 differentiation and IL-4 production, with the latter effect being also present upon fentanyl, buprenorphine, and methadone exposure [15,16]. Such differences may be ligand-dependent [16]. At the same time, also a dose-dependent effect has been demonstrated with different opioids [17]. Looking at T cell activation, morphine has been also shown to reduce major histocompatibility complex (MHC) class II expression, especially on B-cells, leading to CD4+ cell activation and proliferation inhibition [15].

In this systematic review, we aimed at describing the relationship between the immune system and opioids in patients affected by solid tumours.

## 2. Materials and Methods

A systematic literature review was performed by using the following search engines: PubMed, Google Scholar, Cochrane, and Cinahl. The following keywords were used: “opioids” OR “concomitant treatments” AND “neoplasm” OR “tumour” OR “cancer” AND “immunotherapy” OR “immune checkpoints inhibitors” OR “PD-1/PD-L1 inhibitors”. We considered reports published from 1st January 2000 to 1st August 2022. It was decided to include only studies, which analysed adult patients affected by solid tumours. Other inclusion criteria were as follows: (1) the description of ICIs efficacy outcomes in terms of progression-free survival (PFS), overall survival (OS), overall response rate (ORR), time to treatment failure (TTF); (2) concomitant ICIs-opioids treatment. Otherwise, (1) language different than English and (2) not pertinent works were excluded. Two authors (MC and PB) reviewed all the studies and approved the selection following the inclusion and exclusion criteria. In case of disagreement a third author (LB) was asked to make the final decision. The following outcomes considered were: PFS, OS, ORR and TTF. Finally, the reviewing process followed PRISMA guidelines [18] even if the authors did not provide its registration to PRIMA website.

## 3. Results

Five hundred and twenty-three studies were analysed, including abstracts, posters, and oral presentations for international meetings [such as the European Society of Medical Oncology (ESMO) and the American Society of Clinical Oncology (ASCO) meetings]. Thirteen studies were finally selected (Figure 1). Of these, four studies were presented at international congresses, one is a preprint report, while the others are published. The studies were conducted from September 2014 to July 2021. The studies’ characteristics are summarised in Table 1. All the studies of our series were designed as retrospective collections of clinical data. Among these, 5 out of 13 involved more than one centre. Collectively, data about patients with more than eight different tumour types were enrolled, with non-small-cell lung cancer (NSCLC) being the most frequent. Other tumours included were melanoma, Merkel cell carcinoma, renal cell and urothelial cancers, head/neck tumours, colon, and gynaecological cancers. One study did not specify the type of tumours [19]. The studies included in our series were characterised by different sample sizes (range 64–1012) [3,20]. Although treatment with ICIs was an inclusion criterium for all the studies, only in 8 out of 13 studies the type of ICIs was specified. While nivolumab was the most adopted ICI, others included as follows: pembrolizumab, atezolizumab, ipilimumab (combined with nivolumab), and avelumab. Kostine et al. only specified the main targets of the ICIs included in their series (i.e., PD-1/PDL-1 or CTLA-4) [21]. In almost all studies, ICIs were used both as first-line treatments and as subsequent therapies. Only one series included patients treated as first-line treatment only [20]. Oral opioids can be divided into strong opioids (morphine, hydrocodone, oxymorphone, oxycodone, fentanyl, buprenorphine, tapentadol, methadone, and hydromorphone) and weak opioids (tramadol and codeine). In this regard, 5 out of 13 studies reported the type of prescribed opioids. In particular, Botticelli et al. included only patients treated with strong opioids (oxycodone, morphine, fentanyl), while Kostine et al. patients who used only morphine [21,22]. Taniguchi and colleagues reported data about specific administered molecules (i.e., fentanyl, morphine, hydromorphone, tapentadol, and combined oxycodone-fentanyl), while Mock and colleagues divided the patients into low and high opioids users based on a morphine equivalent daily dose (MEDD; <50 and >50 mg, respectively) [23,24]. Similarly, Weinfeld divided patients into the following three groups: opioids-naïve, those treated with low-dose or high-dose of opioids, considering 60 morphine milligram equivalents per day as a cut-off [19].

Studies were heterogeneous also when accounting for main outcomes, including progression-free survival (PFS), overall survival (OS) and in some cases, time to treatment failure (TTF) or response rate (RR). However, all reports showed a negative correlation in terms of OS, PFS or TTF between opioids and ICIs. Taniguchi et al., by analyzing 38 patients treated with opioids matched with others 38 opioids naïve, showed a decreased mOS for the first group (4.20, 95% CI 2.53 to 6.20 months, vs. 9.57, 95% CI 2.23 to not reached months; *p* = 0.018) [23]. Similarly, in 167 patients treated with opioids, the overall response rate was compared to not-treated patients (16.2% vs. 33.7%; *p* < 0.001) [25]. Another study analyzed the concomitant use of opioids in 64 cases of advanced NSCLC treated with single-agent ICIs in first-line setting showing a reduced median progression-free survival (PFS) for patients treated with concomitant opioids as compared to those not receiving opioids (1.7 months versus 12.7 months, HR 4.16, 95%CI 2.15–8.05, *p* < 0.001). These associations were maintained at a multivariate analysis that included performance status, clinical stage, and a number of metastatic sites [20]. The following results were obtained by Miura et al.: at the multivariate analyses, opioid therapy was associated with a shorter OS (HR 1.54; 95% CI: 1.12–2.11, *p* = 0.007), together with subsequent lines of treatment and higher ECOG PS [26]. Other studies confirmed this association, especially with high doses of opioids. Indeed, a retrospective study of 212 patients showed a significantly shorter mOS in patients receiving high doses of opioids (at least 60 morphine milliequivalents daily) as compared to those who received low doses (less than 60 morphine milliequivalents daily) or were opioids naïve (mOS 10 vs. 18 vs. 37 months; *p* = 0.0515) [19]. Similar results were observed by Mock et al. in NSCLC patients treated with high versus low dose opioids therapy (MEDD respectively of >50 and <50 and mOS 3.8 vs. 14.5 months, *p* = 0.001) [24].

**Table 1 healthcare-11-00116-t001:** Characteristics of the selected studies.

Source	Type of the Study	Inclusion Criteria	Other Drugs Than Opioids Evaluated	N. of Patients	Type of Opioids Evaluated	Type of Cancer	Type of ICIs	Results
Bironzo * [20]	Retrospective multicentric observational study	- Advanced NSCLC; - First line treatment with ICIs.	No	64	Strong opioids (NOS): mean daily dose equal to 59 mg of oral controlled-release morphine.	- NSCLC: 64	- NOS:64	Median OS and PFS were shorter in the group of patients treated with opioids.
Iglesias-Santamaria [27]	Retrospective multicentre observational study.	- Advanced/stage IV cancer; - ICIs treatment (at least three doses).	Yes	102	NOS	- NSCLC: 56 - Melanoma: 10 - RCC: 12 - Bladder: 11 - Head and neck: 10 - Others: 3	- Ipilimumab and Nivolumab: 1 - Nivolumab: 61 - Pembrolizumab: 25 - Atezolizumab: 15	PFS and OS were shorter in patients treated with concomitant opioids.
Cortellini [3]	Retrospective multicentric observational study.	- Stage IV cancer; - ICIs treatment.	Yes	1012	NOS	- NSCLC: 528 - Melanoma: 263 - RCC: 185 - Others: 36	- Pembrolizumab: 343 - Nivolumab: 613 - Atezolizumab: 32 - Other: 24	Reduced PFS and OS in patients treated with opioids: higher risk of disease progression and death.
Taniguchi [23]	Retrospective monocentric observational study.	- Advanced NSCLC; - Nivolumab at any line of therapy.	No	296	Oxycodone, fentanyl, morphine, hydromorphone, tapentadol, oxycodone-fentanyl.	- NSCLC: 296	- Nivolumab: 296	PFS and OS were shorter in patients treated with concomitant opioids.
Botticelli [22]	Retrospective multicentric observational study.	- Stage IV cancer; - ICIs treatment.	Yes	193	Strong opioids: morphine, fentanyl and oxycodone.	- NSCLC: 59 - Melanoma: 99 - RCC: 28 - Urothelial cancer: 5 - Merkel carcinoma: 1 - Colon cancer: 1	- Nivolumab: 121 - Pembrolizumab: 60 - Atezolizumab: 11 - Avelumab: 1	PFS and OS were shorter in patients taking opioids as concomitant drugs.
Gaucher [25]	Retrospective monocentric observational study.	- Stage IV cancer; - ICIs treatment.	Yes	372	NOS	- NSCLC: 166 - Melanoma: 110 - Urothelial and RCC: 27 - Head and neck: 48 - Blood cancer: 5 - Gastrointestinal: 4 - -Others: 12	- Pembrolizumab: 130 - Nivolumab: 217 - Ipilimumab: 15 - Ipilimumab and Nivolumab: 10	OS and response rate resulted reduced in patients treated with concomitant opioids.
Kostine [21]	Retrospective monocentric observational study.	- Advanced cancer; - ICIs treatment.	Yes	635	Strong opioids: morphine	- NSCLC: 150 - Melanoma: 293 - RCC: 83 - Utothelial cancer: 16 - Head and neck: 48 - Blood cancer: 20 - Gastrointestinal liver: 16 - Others: 9	- Anti-CTLA4: 3 - Anti-PD-1: 435 - Anti-PD-L1: 66 - -Sequential CPI: 100 - Combined: 31	Median OS and PFS were shorter in the group of patients treated with opioids.
Miura [26]	Retrospective monocentric observational study.	- Advanced NSCLC; - ICIs treatment.	Yes	300	NOS	- NSCLC: 300	- Pembrolizumab: 97 - Nivolumab: 203	TTF and OS were shorter in patients treated with opioids.
Mock * [24]	Retrospective monocentric observational study.	- Advanced NSCLC; - ICIs treatment.	No	208	Strong opioids. A difference between high opioid users (MEDD > 50) and low opioid users (MEED < 50) was evaluated.	- NSCLC: 208	- NOS: 208	Reduced OS according to opioids’ dose implied and duration of treatment.
Verschueren [28]	Retrospectivemulticentric observational study.	- Stage IV NSCLC; - ICIs or chemotherapy treatment.	Yes	442	NOS	- NSCLC: 442	- Pembrolizumab: 121 - Nivolumab: 93 - Atezolizumab: 7	Negative influence on OS (even not statistically significant) in both groups.
Varghese * [29]	Retrospective monocentric observational study.	- Advanced NSCLC; - ICIs and opioids treatment;	Yes	869	NOS. A difference between occasional (OME < 120) and chronic users (OME > 240) was evaluated. The authors considered this work noteworthy and it was included in this series, but as the results are not clearly explained, were not mentioned in the main text.	- Gynaecological cancer: 100 - Soft tissue/melanoma: 123 - Thoracic cancer: 337 - Urologic cancer: 103	- NOS: 869	Decreased OS and PFS in patients enrolled.
Weinfeld * [19]	Retrospective monocentric observational study.	- Advanced cancer; - ICIs treatment.	No	212	NOS. Patients divided in low-dose opioids (<60 morphine milliequivalents/day), high-dose opioids (>60 morphine milliequivalents/day) and naïve users.	- NOS: 212	- NOS: 212	Shorter OS in patients treated with opioids.
Yu # [30]	Retrospective monocentric observational study.	- Advanced NSCLC; - ICIs treatment.	No	132	NOS	- NSCLC: 132	- NOS: 132	Reduced OS and PFS compared to not opioids users.

* Abstract, poster, or oral presentation published for international meetings; # preprint report; NSCLC: non-small cell lung cancer; ICIs: immune checkpoint inhibitors; NOS: not otherwise specified; OS: overall survival; PFS: progression-free survival; RCC: renal cell carcinoma; PSM: propensity score matching; TTF: time to treatment failure; MEED: morphine equivalent daily dose. OME: oral morphine equivalent.

## 4. Discussion

Up to 90% of cancer patients experience pain at some stage of their disease journey, with a third rating the intensity of their pain as moderate to severe, and up to half being undertreated [31]. In the management of cancer pain, the prescribed opioids are divided as discussed above into strong and weak. However, weak opioids hold a controversial role in the management of cancer pain and have been demonstrated inferior to low-dose morphine for treating moderate cancer pain [32]. The most common non-opioid drugs prescribed for the treatment of cancer pain are acetaminophen/paracetamol; corticosteroids; non-steroidal anti-inflammatory drugs (NSAIDs); anti-neuropathic agents, which include tricyclic antidepressants and anticonvulsants and, in the end, bisphosphonates. The choice of drug is often driven by the relevance of potential adverse effects (AEs) in the single patient. For example, the toxicity profile for NSAIDs includes gastrointestinal and cardiovascular AEs, nephrotoxicity, and hepatotoxicity. A Cochrane review found 10 studies that compared a NSAID with an opioid, 4 found the NSAID to be more effective, whereas 2 studies showed they were less beneficial. Meta-analyses of four of the studies found no significant difference in pain relief but more AEs with the opioid use [33]. Similarly, corticosteroids have multiple potential short- and long-term Aes, including those on the immune response, behaviour, and carbohydrate/protein metabolism [34]. Despite corticosteroids being used widely to treat cancer pain, there is limited evidence for their efficacy [35]. Opioid analgesia is usually inadequate to obtain neuropathic pain relief, and additional medications are required, mainly antidepressant and anticonvulsant drugs. However, these agents are used mostly in combination with opioids [36]. The evidence for chronic use of most of the non-opioid drugs in the treatment of cancer pain remains scarce. Most studies have methodological limitations and lack long-term follow-up, so that data on the efficacy of use of these drugs remain limited. Some relevant issues persist, for example, whether NSAIDs and corticosteroids can be safely continued long-term in cancer patients, or which non-opioid drugs are best for specific types of pain and in which combinations [31]. A recent systematic literature review stated that evidence on the efficacy and safety of non-opioid drug combinations in the treatment of cancer pain is scant, as few RCTs have been published to date [37]. There is certainly a need to evaluate non-opioid drug combinations in the management of cancer pain. However, further research on this topic is needed to recommend non-opioid drugs to replace or reduce the prescription of opioids for the treatment of cancer pain. Moreover, reduced opioid access could worsen the problem of cancer pain under treatment and threaten decades of progress in the care of patients with advanced cancer. As discussed above, despite the widely spread of ICIs, data about the potential pharmacological interactions of these drugs have been studied only recently. In our systematic review, we collect and summarize the existing evidence between the concomitant use of opioids and ICIs.

Regarding the possible influence of opioids on ICIs efficacy, our systematic review suggests that opioid use may be associated with worse outcomes. None of the studies evaluated specific safety issues when dealing with the concomitant administration of opioids and ICIs. As already stated, our study has several limitations. One of these consists of the retrospective nature of all the included studies. Therefore, concomitant medications have been extracted from prescription files, and some authors did not report the type and/or dosage of opioids. Furthermore, 7 out of 13 studies were designed to include all concomitant drugs during ICIs treatment. Another major limitation bias is the heterogeneity of included population. Indeed, 6 studies enrolled patients affected by different tumours, while 11 studies the included patients treated with different ICIs in different treatment lines. 

In almost all the studies both the performance status and the tumour burden were assessed and evaluated in multivariate analyses, generally maintaining statistical significance, with the exception of the study of Gaucher et al. [25]. It should be emphasised in this regard, as pointed out by Cortellini et al. in their multicentre observational retrospective study, that opioid use at baseline could be associated with lower ECOG-PS and higher tumour burden and may therefore represent another confounding factor [3]. Opioids are usually prescribed to treat and relieve pain, which could be associated with advanced and/or progressive disease. This, as suggested by Miura et al. could explain the association with reduced TTF and OS observed in these patients [26].

Moreover, poli-pharmacy treatments identify patients characterised by several comorbidities, a higher tumour burden and already treated with several lines of treatments, with a well-known reduced response to ICIs [22]. In the last years, a new hypothesis emerged, and it is represented by interrupting the gut microbiome composition leads to disruption of gut homeostasis and the whole immune system. On the other side, we now have robust evidence that microbial flora plays a crucial role in modulating ICIs efficacy by influencing the tumour microenvironment. At least part of ICIs failure may be attributed to specific patients’ microbiome, which shows great variability through individuals and its composition can be influenced by many factors, such as specific drugs used. While antibiotics have been demonstrated to negatively influence gut microbial flora, even opioid use can be called into question. If we consider, together, both statements according to which, on one side, opioids can cause an alteration in gut homeostasis [38], and, on the other side, the gut microbiome is able to influence ICIs efficacy and sensibility [39], we understand that all this has an important value for everyday clinical practice. Today, with the entry into the therapeutic algorithm of the main advanced solid tumours of ICIs and with the parallel development of simultaneous care the interaction between opioids-microbiome-ICIs assumes a non-negligible dimension. Everyday clinical practice has to face many healthcare needs, especially for cancer patients. In the last years, the widely spread of immunotherapy thanks to its promising results, has led to new possible treatments also for patients affected by NSCLC. The availability of these new therapeutic approaches made it possible to focus not only on patients’ prognosis but also on their quality of life. In this regard as stated by international guidelines, pain control should be always evaluated and achieved. In NSCLC patients, it can depend on predominantly bone metastases, which regard almost 30–40% of patients [40] and due to its characteristics and intensity, most patients are treated with oral opioids from the earliest phases of the disease. However, as described above, recent preclinical [4] and clinical evidence may raise doubts about the use of opioids in patients treated with ICIs due to possible interactions. The results achieved from different studies are based on retrospective and heterogeneous data and therefore not proper to describe the phenomenon effectively. Assuming an established influence of opioids in the case of treatment with ICIs, it should be stated if it regards every level of dose prescribed or whether there is an equal level of safety for all patients. These doubts are moreover legitimate considering the results of the works of Weinfeld and Mock, who found differences in PFS considering various doses of opioids in NSCLC patients treated with ICIs [19,24]. Pain relief can be achieved also using palliative radiotherapy (RT), which is often considered due to good results and generally high tolerability. Moreover, recent studies showed a synergistic action between RT and ICIs due to for example the depletion of regulatory T cells by RT in the tumour microenvironment [23,41]. Due to several drugs, which can influence ICIs efficacy, such as antibiotics, corticosteroids, PPIs and in a less clear way, opioids [21], it should be encouraged a mindful use of concomitant therapies during ICIs treatment in terms of timing and dosage, evaluating therapeutic appropriateness and real utility.

## 5. Conclusions

Oral opioids seem to impact ICIs efficacy in cancer patients in different and not completely known ways. Possible mechanisms rely on the presence of morphine receptors on cancer cells and the influence of opioids on the immune system; others consider the role of the microbiome. To better understand the relationship between ICIs and concomitant drugs such as opioids, prospective studies with large sample sizes should be encouraged. Concomitant drugs during chemotherapy and ICIs treatment should be carefully prescribed even if, as of today, there is not enough evidence to avoid the prescription of opioids in patients treated with ICIs.

## Figures and Tables

**Figure 1 healthcare-11-00116-f001:**
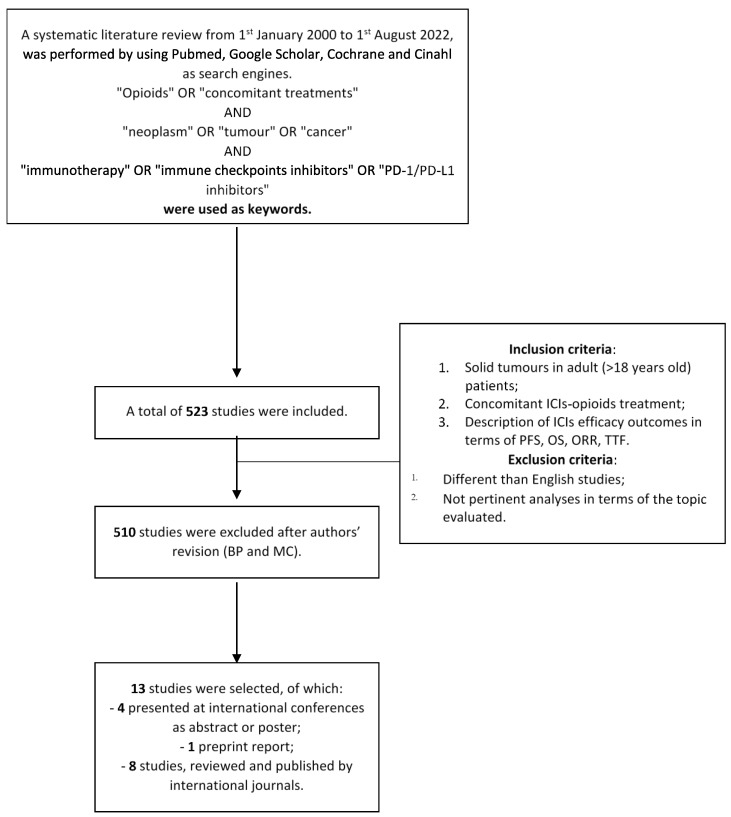
Brief design of the systematic review.

## Data Availability

Not applicable.

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
