# Peer review of "Immune Checkpoint Inhibitors and Opioids in Patients with Solid Tumours: Is Their Association Safe? A Systematic Literature Review"

_healthcare, 2022, doi:10.3390/healthcare11010116_

Round 1

Reviewer 1 Report

The manuscript “Immune checkpoint inhibitors and opioids in patients with cancer: is their association safe? A systematic literature review”raised up a thought-provoking question. As immune checkpoint inhibitors expand in the treatment of a widening range of malignancies, and majority of cancer patients assume concomitant medications for the treatment ,of which the interactions have become an area of increasing interest due to the potential  influence on the outcomes from immunotherapy. The author conducted a systematic review and drawed the conclusion. However, there exists some questions in this review.

1. The systematic review should have been conducted in accordance with Meta-analyses (PRISMA) guidelines and have been registered in the International Prospective Register of Systematic Reviews.

2. The word Cancer in the title of the review is not specific, which might be substituted for the solid tumors as it mentioned in the Methods Part.

3. There are more than 8 different tumor types enrolled in the study ,of which there is insufficient numbers. Combining of them will result in inevitable bias.

4. The author mentioned that they just reviewed the results instead of the titles and abstracts of all potential studies independently when they make a selection, so it is uncertain whether the ineligible and qualified studies were included.

5. Data extraction and statistical analysis were not carried out and no reasons were given for that.

6. The author left out the period at the end of the Materials and Methods paragraph.

Author Response

On behalf of all the authors, I would like to sincerely thank You for the revisions and comments to our work. Your observations and suggestions allowed us to reconsider parts of our paper in order to make it more complete.

We hope having solved all the critical aspects you pointed out.

In particular, for what regards Rev_1:

Q1. The systematic review should have been conducted in accordance with Meta-analyses (PRISMA) guidelines and have been registered in the International Prospective Register of Systematic Reviews.

R1: We thank Rev_1 for this comment. Even if not specified, we conducted our analysis according to PRISMA guidelines. As this was not highlighted in the main text, we decided to make it clearer, specifying all the required statements pointed out in PRISMA guidelines. As not mandatory, we decided not to register our systematic revision in the International Prospective Register of Systematic Reviews.

Q2. The word Cancer in the title of the review is not specific, which might be substituted for the solid tumors as it mentioned in the Methods Part.

R2: In order to avoid any confusion in readers who might take in consideration our revision in search engines or databases considering only the title, we decided to change it as follow: “Immune checkpoint inhibitors and opioids in patients with solid tumours: is their association safe? A systematic literature review”.

Q3. There are more than 8 different tumor types enrolled in the study, of which there is insufficient numbers. Combining of them will result in inevitable bias.

R3: We thank Rev_1 for this observation. During the revision process, we met several difficulties in finding pertinent works to the topic of our review. Moreover, most analyses were based on retrospective data collections including several different tumor types. We acknowledge this important bias which is related to the current available literature on this topic. Therefore, insufficient heterogeneous numbers are an unavoidable expected limit of our work and combing them, as suggested by Rev_1, would have resulted in inevitable bias. We state this in our conclusions.

Q4. The author mentioned that they just reviewed the results instead of the titles and abstracts of all potential studies independently when they make a selection, so it is uncertain whether the ineligible and qualified studies were included.

R4: We thank Rev_1. During the revision process, PB and MC analysed not only the results of the studies but all the paper, including title, abstract and the main text (when available). The word “results” in the text was a mistake that we provided to solve.

Q5. Data extraction and statistical analysis were not carried out and no reasons were given for that.

R5: Considering the results obtained and the type of included studies, we decided not to perform statistical analyses due to the heterogeneity of the data included.  

Q6. The author left out the period at the end of the Materials and Methods paragraph

R6: We thank Rev_1 for this observation. We provided to solve it.

Reviewer 2 Report

By reviewing several retrospective multicenter observational studies, the authors sought to identify the role of opioids in patients with solid tumors receiving ICIs. There are several issues that need to be addressed before this review can be published in Healthcare Journal.

Q1. It is very helpful for the author to elaborate on the mechanism of opioids on the immune system or immune cells in the Introduction.

Q2. Could the authors add opioids to the table for each study so that it would be easier for readers to understand the therapeutic impact of each opioid on ICI.

Q3. I am very curious that the author chose five hundred and twenty-three studies at the beginning, but finally chose the thirteen study for analysis. What is the basis for this choice? Can the author please clarify.

Author Response

On behalf of all the authors, I would like to sincerely thank You for the revisions and comments to our work. Your observations and suggestions allowed us to reconsider parts of our paper in order to make it more complete.

We hope having solved all the critical aspects you pointed out.

By reviewing our work, Rev_2 pointed out three different critical issues. We thank the Reviewer for his work, in particular:

Q1. It is very helpful for the author to elaborate on the mechanism of opioids on the immune system or immune cells in the Introduction.

R1: We thank Rev_2 for his suggestion. We agree with him and we provided to modify the main text as suggested trying to elaborate this part of the paper in order to draw a more complete background of our study.

Q2. Could the authors add opioids to the table for each study so that it would be easier for readers to understand the therapeutic impact of each opioid on ICI.

R2: We thank Rev_2 for this advice. Having added opioids type (when available) to table_1, made the work more complete.  

Q3. I am very curious that the author chose five hundred and twenty-three studies at the beginning, but finally chose the thirteen study for analysis. What is the basis for this choice? Can the author please clarify.

R3: We thank Rev_2 for this comment. Five hundred and twenty-three studies were the results of our preliminary research. Filtering these studies considering the inclusion and exclusion criteria, MC and PB selected at the end 13 studies. More specifically, authors decided to exclude all not pertinent studies, including those which did not evaluate ICIs efficacy outcomes related to opioids treatment or those that considered opioids just in terms of drug-addiction. In the end, all studies that evaluated concomitant therapies other than opioids were not considered. The main text did not explicit the inclusion and exclusion criteria and so we provided to solve this limitation.

Round 2

Reviewer 1 Report

The author has revised and elaborated according to the revision. Despite the lack of evidence for inclusion in the study, the issue was elaborated as much as possible in the context of the available studies.

Please note that there is a spelling error of 'complexly’ in the last sentence of the RESULTS, please correct it promptly.

Author Response

1. We thank Rev_1 for this comment. We provided to correct it.

Reviewer 2 Report

The author responded well to questions about the article, which helps readers understand the article better. I agree to publish this article. Thanks for the author's work.

Author Response

We thank Rev_2 for his work.
